# Tongue strength and endurance among typically developing children and children with idiopathic speech sound disorders in the United Arab Emirates

**Mohammed Safi**[1], **Dalia Mohammad Alzyod**[2], **Maxwell Peprah Opoku**[3]*, **Yasser E. Agamy**[4]

1 Department of Speech Language Pathology, College of Medicine and Health Sciences, United Arab Emirates University, Al Ain, UAE, 2 Speech Language Pathologist, United Arab Emirates University, Al Ain, UAE, 3 Special Education Department, College of Education, United Arab Emirates University, Al Ain, UAE, 4 General Physician and Clinical Tutor, Thumbay University Hospital Complex, Ajman, United Arab Emirates

* maxwell.p@uaeu.ac.ae

**Data Availability Statement:** Data cannot be shared publicly because of ethical restrictions. Data are available from the United Arab Emirates University Ethics Committee (contact via research.

## Abstract

### Background

Tongue strength and endurance in adults have been extensively studied, but data on these parameters in young children remain largely unavailable.

### Aims

This study aimed to collect normative objective tongue strength and endurance data from a pediatric population in the United Arab Emirates (UAE) and to examine the effects of age and sex on these parameters.

### Methods

This normative study included a total of 65 typically developing (TD; n = 36) children and children with idiopathic speech sound disorders (ISSDs; n = 29). The participants were assigned to four age groups (range: 3–8 years) and stratified by sex. Data on their tongue strength and endurance were collected using the Iowa Oral Performance Instrument.

### Results

Tongue strength scores and endurance time increased with age in both the TD and ISSD groups. Sex had no statistically significant effects on tongue strength or endurance in either group. Furthermore, tongue strength and endurance time scores were higher in the TD group than in the ISSD group.

### Conclusions

The data obtained in this study would add important normative data to the database of standardized measurements for maximal strength and endurance scores in the pediatric

office@uaeu.ac.ae) for researchers who meet the
criteria for access to confidential data.

**Funding:** The author(s) received no specific
funding for this work.

**Competing interests:** The authors have declared
that no competing interests exist.

population of the UAE. Future research is encouraged to collect additional data that can
help healthcare professionals objectively evaluate children with feeding, swallowing, and
speech sound production difficulties.

## Introduction

The tongue is an important muscle that plays a dual role in swallowing and speech mechanisms. Anatomically, it consists of two muscle groups (intrinsic and extrinsic); precise changes in their positions and pressure accomplish different functions [1]. Normal swallowing requires the maintenance of adequate tongue pressure and endurance [2]. Consequently, the assessment of tongue strength and endurance and their effects on swallowing and speech is an important component of the comprehensive evaluation of speech and language [3]. Objective lingual muscle strength measurement is also an essential addition to clinical evaluation and should replace subjective measurements [4]. Therefore, there has been a recent movement in clinical research toward the development of objective tongue strength measurement tools [4]. Unfortunately, studies on tongue strength and endurance are rare in the Middle East.

Several devices are used by clinicians to measure tongue strength data and facilitate therapy planning and training. For instance, SwallowSTRONG (Swallow Solutions, LLC, Madison, WI, USA) uses sensors in the mouthpiece to measure tongue pressure at four locations [5, 6]. Similarly, Tongueometer™ (E2 Scientific Corp., Walnut, CA, USA) is a device that offers four modules for assessing and increasing tongue strength and endurance, in addition to providing easy data and report access through mobile applications [7]. Another device is the Iowa Oral Performance Instrument (IOPI®), which is used to measure tongue strength, tongue endurance, and lips is measured by recording the maximum pressure produced by an individual via pressing an air-filled bulb against the hard palate with the tongue [8, 9].

The IOPI® has been used successfully in studies on oral phase swallowing function in dysphagic populations [9] and is considered a standard tool for tongue pressure measurements and lingual muscle-strengthening interventions [10]. It reportedly has good inter-rater reliability [11] and test–retest reliability [12] for tongue endurance. In a more recent study, Potter et al. [13] reported high intrasubject test–retest reliability ($r = 0.89$) in the control group during evaluation. This probably makes the IOPI a useful tool for studying tongue strength and endurance among children in a novel context, such as the UAE. Although there are abundant data related to tongue strength and endurance in the adult population [9, 14, 15], data on these parameters remain limited in the pediatric population [15–20].

Demographic differences in muscle strength exist. In fact, the effects of child characteristics, such as age and gender, on skeletal muscle strength have been well established [21]. Although inconclusive, findings on the effects of demographic variables on tongue strength have been found to be nonstatic [21–25]. For instance, replication studies comparing the tongue strength data of English speakers with those of Portuguese, French, and Korean speakers have found cultural differences. For instance, Vitorino [25] reported that healthy Portuguese speakers had lower tongue endurance scores compared with English speakers. Jeong et al. [22] compared the tongue strength data of a Korean population with those of American and Belgian populations and found a difference of 15 kPa between the three populations; however, the authors indicated that this difference was caused by an error in measurement rather than a cultural difference. In another study, Vanderwegen [23] found a difference in tongue strength among Belgian children, with the comparison revealing statistical differences across age. Comparatively,

contributions from the Arabic world regarding tongue strength and endurance are evidently unexplored.

Overall, these findings suggest the need to further investigate the nature of such differences not only in healthy individuals but also in children with idiopathic speech sound disorders (ISSDs). These disorders include those associated with the motor production of speech sounds and the linguistic aspects of speech production [26]. Articulation disorders focus on errors (distortions and substitutions) in the production of individual speech sounds. Although investigators continue to propose novel and innovative ideas that target lingual muscle strength measurements [10], there is limited evidence of the measurement of tongue strength and endurance in non–English-speaking populations. Collecting such normative objective data from children will provide speech–language pathologists (SLPs) and other allied health professionals with valuable information on the following aspects of tongue function: evaluation, diagnosis, and planning of relevant interventions. Furthermore, standardized tongue strength data have not been collected from regions such as the United Arab Emirates (UAE) and the Middle East.

This study was designed to measure tongue strength and endurance in both healthy children and those with ISSDs in the UAE using the IOPI®. The IOPI® was used to collect normative data on tongue strength and endurance in a sample of typically developing (TD) Arabic-speaking children and a sample of children with ISSDs. The effects of age and sex on these parameters were considered in both groups. The following research questions were addressed:

1. What are the tongue strength and endurance levels of TD children and children with ISSDs in the Arabian context?

2. Are there differences between participants across child type (TD vs. ISSDs), sex, and age on tongue strength and tongue endurance?

3. Are there associations between child type (TD vs. ISSDs), age, and sex?

## Methods

### Study design

This study was a case–control study of young children who were TD and those with ISSDs. It was approved by the UAE University Ethics Committee (approval no. ERS_2021_7249). The parents of each participant signed a written consent form to allow their children to participate in the study.

### Participants

All participants in this study (a) were between 3 and 7 years old, (b) spoke Arabic as their first language, (c) had no hearing loss based on hearing screening, and (d) had no chronic diseases or disabilities, with the exception of articulation disorder in those children who were included in the ISSD group. They were assessed by a licensed SLP using (a) oral motor examination, (b) articulation screening, and (c) data obtained from the medical and speech and language case history forms filled out by their parents. The SLPs involved in the assessment were blinded to the aims of this study to increase its validity. Table 1 presents the demographic characteristics of the participants.

Two groups of children were recruited for this study: TD (n = 36) and those with ISSDs (n = 29). Most of the children were enrolled in preschools or schools. Thirty-nine TD children

**Table 1. Summary of tongue strength and endurance data.**

| Category | Frequency (N = 65) | Tongue strength | Tongue endurance |
|---|---|---|---|
| **Type of child** | | 36.25 (11.91) | 10.00 (4.71) |
| Children with ISSD | 29 (44%) | 27.79 (8.00) | 9.17 (4.10) |
| Typically developing children | 36 (56%) | 43.06 (10.05) | 10.67 (5.11) |
| **Sex** | | 36.25 (11.91) | 10.00 (4.71) |
| Boys | 35 (54%) | 37.40 (12.40) | 9.97 (4.95) |
| Girls | 30 (46%) | 34.90 (11.36) | 10.03 (4.50) |
| **Age** | | 36.25 (11.91) | 10.00 (4.71) |
| 3–5 years | 28 (43%) | 31.07 (10.39) | 6.04 (1.62) |
| 6–8 years | 37 (57%) | 40.16 (11.59) | 13.00 (4.00) |

were recruited to participate in this study, but three of them did not cooperate during the data collection sessions, so they were not able to complete the protocol. None of them had a history of cleft lip/palate, neurodevelopmental disorders, feeding difficulties, swallowing difficulties, motor planning speech disorders (apraxia of speech and dysarthria), dental deformities (mal-occlusions), neurological impairments (cerebral palsy, stroke, and traumatic brain injury), hearing loss, intellectual deficits, receptive–expressive language delay, meningitis, fluency disorders, genetic syndromes (Down syndrome and fragile X syndrome), or organic speech sound disorders (of motor/neurological, structural, or sensory/perceptual causes).

Conversely, 29 children with ISSDs were involved in this study. The presence of ISSDs was confirmed via an evaluation performed by a certified SLP. The children with ISSDs were enrolled in or had undergone speech therapy. They exhibited consistent speech sound errors, with little to no variation across words and sittings [5].

After recruiting 65 participants, the research team deemed the sample appropriate based on the initial sample size estimation (http://www.openepi.com/SampleSize/SSCC.htm). The sample was estimated using previous study findings reporting that an estimated 18% of the US population could be living with a form of ISSD, whereas 82% were found to be free of any ISSDs [27]. With this in mind, our parameters for the computation were follows:

$$\text{Confidence interval} = 99\%$$

$$\text{Power} = 80\%$$

$$\text{Ratio of the controls to the case} = 2$$

$$\text{Hypothetical proportion of the controls with exposure} = 18\%$$

$$\text{Hypothetical proportion of the cases with exposure} = 82\%$$

With the use of Fleiss continuity correction, it was determined that a minimum of 35 participants were required for this study. The ratio was 12 cases to 23 controls. However, in this study, the minimum target was exceeded because 65 participants (29 cases: 36 controls) were recruited. Furthermore, as stated earlier, the control group originally consisted of 39 children, but three did not cooperate during the data collection. Nevertheless, the minimum sample size was still exceeded.

## Procedures

The participants in this group were recruited from among the populations residing in Dubai and Al Ain, UAE, via email and WhatsApp messages. The participants were obtained from rehabilitation centers, speech therapy clinics, and schools across Dubai and Al Ain, UAE. The research team contacted schools, rehabilitation centers, and clinics for permission to recruit children for the study. The researchers subsequently sent information about the study to parents, inviting them to participate in the research. Parents who wanted their children to join the study were asked to meet with the research team at a given date and time.

During the data collection sessions, the participants were first seated in an upright position on a chair and instructed to minimize their movements during the trials. A picture-naming articulation test was used to evaluate the articulation abilities of the children in this study. This type of test was chosen because it enables the elicitation of all speech sounds of the Arabic language in three word positions. It consisted of 60 pictures targeting 28 consonants in the initial, medial, and final positions.

Data were collected from January 05, 2021, to June 15, 2021. Each data collection session lasted for around 30 min. Each participant was allocated a new sterilized and disposable IOPI Tongue Bulb 5–6010 [28, 29] (Fig 1). Because of the participants' age, verbal motivation was given to encourage their continued participation in the trials.

## Instrumentation

The two study variables—tongue strength and tongue endurance—were measured using the IOPI (IOPI Medical LLC, Carnation, WA, USA) (Fig 1). The procedure used in this study followed the instructions outlined in the IOPI® manual [28]. Several studies that used the IOPI® have followed the manual procedure for conducting measurements [25, 28, 30, 31].

A. Maximum tongue strength. To measure maximum tongue strength, the examiner first demonstrated the placement of the tongue bulb on himself/herself and then placed it for the participants. For the first three training trials, the tongue bulb was placed on the oral part of the tongue dorsum (Fig 2), with the tongue bulb pushing right under the alveolar ridge. For the younger age group (3–6 years old), the examiner held the stem of the bulb to ensure that the placement was accurate. After bulb placement, using their tongues, the participants were instructed to push the bulb against the hard palate as hard as they could for approximately 2 s. After a 2 min break, the participants were instructed to repeat the same procedure for three more trials, with a 60 s rest time between trials. The values were displayed on the IOPI monitor. The highest value obtained in any of the three trials was recorded as the maximal tongue strength in kilopascals.

B. Tongue endurance. The light setting on the IOPI system was activated to measure tongue endurance. The isometric pressure was set to 50% $P_{max}$ of each participant's maximal tongue strength value, recorded from the previous procedure [28]. The bulb placement used in the tongue strength measurement was also applied here (Fig 2). Using their tongues, the participants were instructed to push the bulb against the hard palate as hard as they could for as long as possible to ensure that the green light stayed on as visual feedback. The first trial was dismissed and repeated if the participant could not ensure that the green light stayed on for a minimum of 2 s continuously. The endurance values were timed and recorded in seconds, representing how long the participants could ensure that the green light stayed on. The procedure was repeated three times, with a 60 s rest time between the trials. A 5 min rest was taken between the tongue strength and tongue endurance measurements [29].

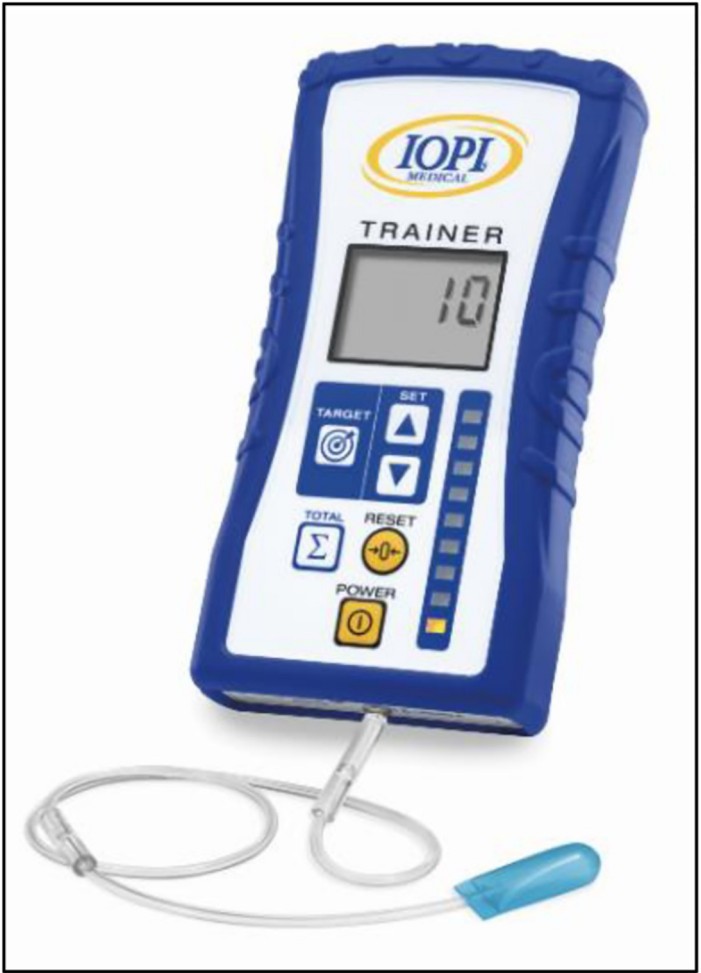

**Fig 1. Iowa Oral Performance Instrument.** The image was taken from IOPI Medical [29].

## Data analysis

The data were entered into Microsoft Excel before being transferred to SPSS for analysis. Owing to the sample size of the study, a non-parametric test was deemed appropriate. The collected data were stratified by sex (male and female) and age (3–5 and 6–8 years) in both the TD and ISSD groups. Furthermore, correlation analysis was performed to ensure the test–retest and inter-rater reliability of the measurements. The intraclass correlation scores were .60 for single measures and .71 for average measures [32]. Consequently, the intraclass correlations could be interpreted as acceptable. Following this, the research questions were answered.

For research question 1, mean scores were calculated to understand tongue strength and endurance in the children who took part in this study. These scores were also computed across sex and age. The higher the mean scores, the greater the tongue strength and endurance.

For research question 2, the Mann–Whitney $U$ test was performed to determine the differences between participants (across child type, sex, and age) in terms of tongue strength and endurance. The effect size was reported as small (.01–.30), medium (.31–49), or large (at least .50) [32].

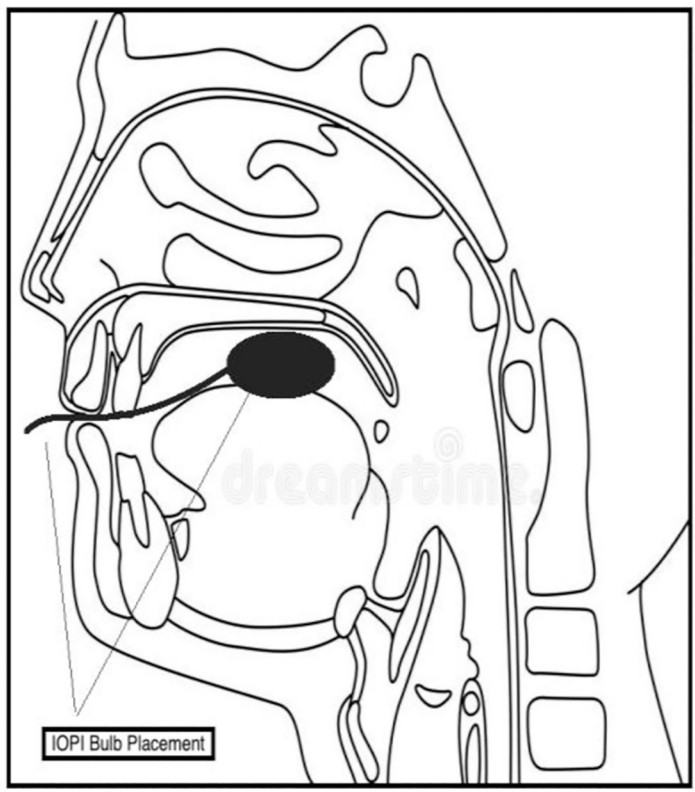

**Fig 2. Iowa Oral Performance Instrument (IOPI) bulb placement.**

To answer research question 3, a chi-square ($\chi^2$) test of independence using Yates' continuity correction was conducted to explore the association between the categorical variables (child type [TD vs. ISSDs], age, and sex) [32]. Statistical significance for the differences between participants was set at a *p* value of < .05. In relation to the effect size, the $\varphi$ coefficient was reported as small (.01–.30), medium (.31–49), or large (at least .50) [32].

## Results

A total of 65 children were recruited for this study. Of these, 56% were TD children, whereas 44% were children with ISSDs. In terms of sex, 54% were boys, and 46% were girls. In terms of age, most of the participants were between the ages of 6 and 8 years.

### Levels of tongue strength and endurance

Mean scores were computed to determine tongue strength and endurance among the children. In terms of child type, tongue strength (*M* = 43.06, standard deviation (*SD*) = 10.67), and endurance (*M* = 10.67, *SD* = 5.11) levels in the TD group were higher than those in the ISSD group (see Table 1). In terms of sex, the mean scores showed that tongue strength was higher in boys (*M* = 37.40, *SD* = 12.40) than in girls (*M* = 34.90, *SD* = 11.36), but endurance was higher in girls (*M* = 10.03, *SD* = 4.50) than in boys (*M* = 9.97, *SD* = 4.95). With respect to age, tongue strength and endurance levels were higher in the older children than in the younger ones.

## Association between background variables: Child type (TD vs. ISSDs), age, and sex

Chi-square $\chi^2$ tests of independence were conducted to explore the relationships between type of child, sex, and age (see Table 2). A Chi-square $\chi^2$ test of independence was first conducted to ascertain the proportion of child type based on sex. According to Yates' continuity correction, the results showed no significant relationship between type of child and sex [$\chi^2_{(1,\ N = 65)}$ = 1.12, $p$ = .29, $\varphi$ = −.16]. Similar results were observed between type of child and age [$\chi^2_{(1,\ N = 65)}$ = .06, $p$ = .89, $\varphi$ = −.03] and between sex and age [$\chi^2_{(1,\ N = 65)}$ = .05, $p$ = .83, $\varphi$ = .06].

## Differences between children's endurance and tongue strength

The differences between the categorical variables in terms of tongue strength and endurance were assessed using the Mann–Whitney $U$ test (see Table 3). Significant differences were observed between the ISSD group ($Mdn$ = 26, $N$ = 65) and the TD group ($Mdn$ = 42, $N$ = 65) in terms of tongue strength only ($U$ = 931.00, $z$ = −1.23, $p$ = .001, $r$ = 0.15) (Fig 3). There were also significant differences between the different age groups in terms of tongue strength and endurance. Specifically, those participants between 6 and 8 years old ($Mdn$ = 39, $N$ = 65) scored higher in tongue strength than those between 3 and 5 years old ($Mdn$ = 33, $N$ = 65) ($U$ = 740, $z$ = −1.23, $p$ = .001, $r$ = 0.15). Similar patterns were observed for tongue endurance, with participants between 6 and 8 years old ($Mdn$ = 13, $N$ = 65) scoring higher than those between 3 and 5 years old ($Mdn$ = 6, $N$ = 65) ($U$ = 980, $z$ = −1.23, $p$ = .001, $r$ = 0.15) (Figs 4 and 5).

## Discussion

While studies on tongue strength are gaining traction in western or advanced societies [30, 31, 33, 34], little attention has been given to this topic in non-Western societies. The present work was a case–control study that investigated tongue strength and endurance in young children in the UAE who were TD and those with ISSDs. We believe that this is the first study to have evaluated normal and abnormal lingual muscular abilities involving participants from the Middle East.

This study showed that tongue strength scores increased with age in the study population. Similarly, the findings showed that age had a significant effect on tongue endurance. The older the participants, the higher their strength and endurance time scores. This finding agrees with

**Table 2. Summary of relationships between categorical variables.**

| Type of child | Boys | | Girls | | $\chi^2$ |
|---|---|---|---|---|---|
| | $n$ | % | $n$ | % | |
| Children with ISSD | 13 | 45 | 16 | 55 | .29 |
| TD children | 22 | 61 | 14 | 39 | |
| **Type of child** | **3–5 y** | | **6–8 y** | | $\chi^2$ |
| | $n$ | % | $n$ | % | |
| Children with ISSD | 12 | 41 | 17 | 59 | .06 |
| TD children | 16 | 44 | 20 | 56 | |
| **Sex** | **3–5 y** | | **6–8 y** | | $\chi^2$ |
| | $n$ | % | $n$ | % | |
| Boys | 16 | 46 | 19 | 54 | .05 |
| Girls | 12 | 40 | 18 | 60 | |

**Table 3. Association between background variables, tongue strength, and tongue endurance.**

| Category | Tongue strength | Tongue endurance |
|---|---|---|
| **Type of child** | | |
| Children with ISSD | 26 | 8 |
| TD children | 42 | 9 |
| Mann–Whitney $U$ | 931.00** | 597.50 |
| Effect size | .15 | .15 |
| **Sex** | | |
| Boys | 38 | 9 |
| Girls | 37.50 | 8.50 |
| Mann–Whitney $U$ | 472.50 | 548.00 |
| Effect size | .15 | .15 |
| **Age** | | |
| 3–5 y | 33 | 6 |
| 6–8 y | 39 | 13 |
| Mann–Whitney $U$ | 740** | 980** |
| Effect size | .15 | .15 |

those of previous studies reporting differences between children in terms of age [23, 31]. This logical result could be associated with the cognitive and psychological maturation of the participants. It is apparent that as children grow, they demonstrate improved tongue strength and endurance. Therefore, clinicians should consider the cognitive and psychological abilities of the pediatric population before making conclusions related to tongue endurance and strength in the UAE.

The findings also showed that boys and girls showed no statistically significant differences in tongue strength and endurance time scores across all ages. It is apparent that boys and girls demonstrate similar characteristics when it comes to tongue strength and endurance. This finding is consistent with that of a previous study on both pediatric and adult populations,

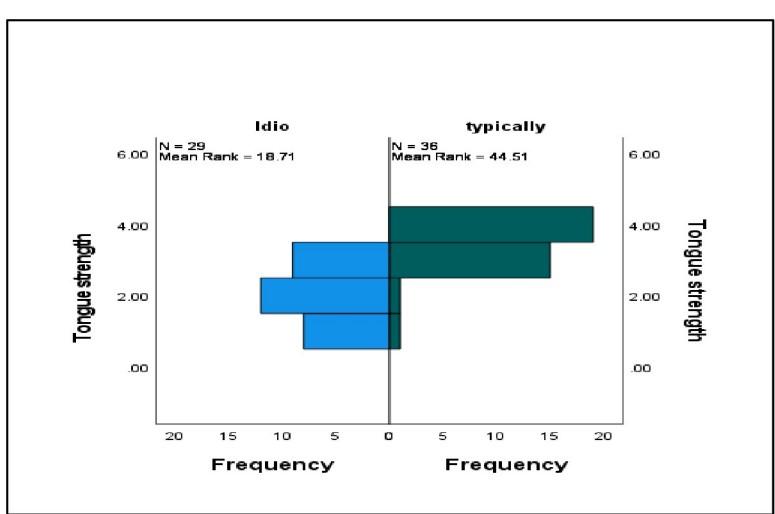

**Fig 3. Comparative data on tongue strength between typically developing children and children with idiopathic speech sound disorders.**

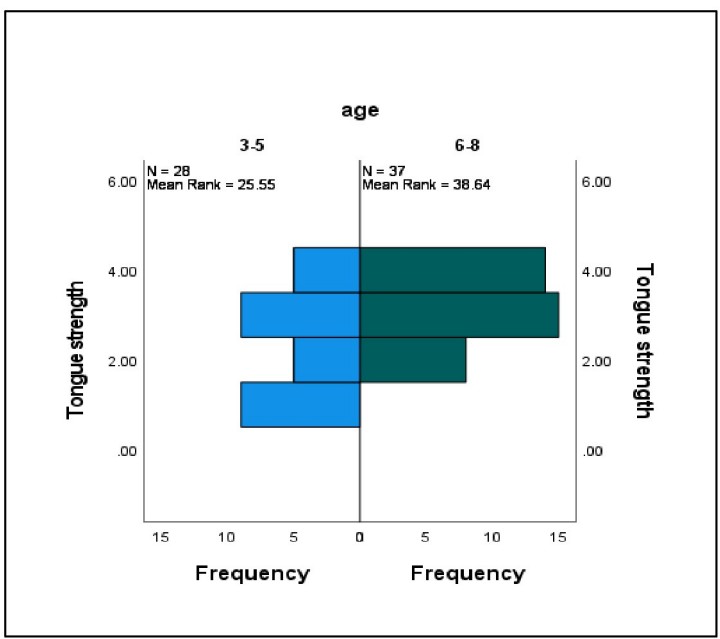

**Fig 4. Comparative data on tongue strength according to age.**

which reported no difference between participants in terms of gender [23–25, 30]. This suggests that clinicians may adopt a common approach when working with boys and girls.

Tongue strength measurements differed significantly between the TD and ISSD groups. The results indicated that the ISSD group had lower tongue strength. Nevertheless, the result is

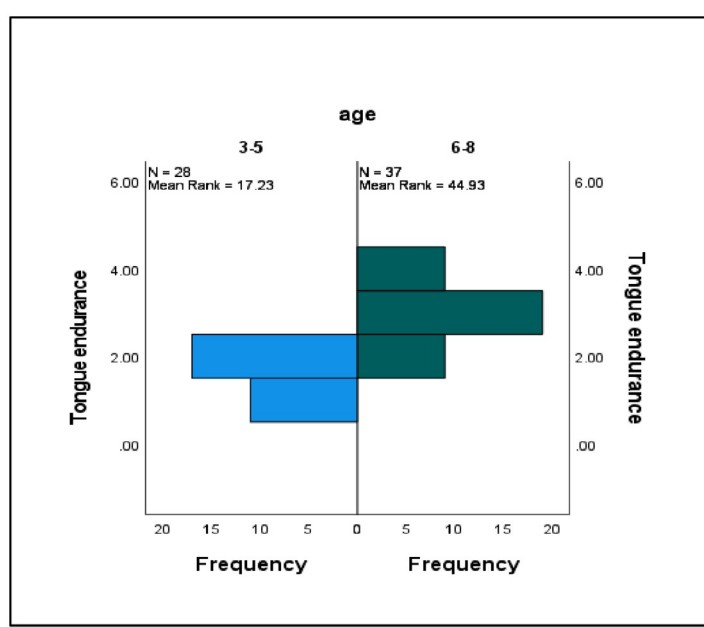

**Fig 5. Comparative data on tongue endurance according to age.**

consistent with a previous study that found a difference between TD children and peers with speech impairment, with the latter demonstrating decreased tongue strength [13, 34]. Clinically, this suggests that some lingual or even oral weaknesses result in difficulty in articulation [13]. However, surprisingly, tongue endurance time did not differ significantly between the two study groups. Notably, during the data collection phase of this study, the children in the younger age group (3–4 years) required a frequent demonstration and explanation of the procedure for the purpose of the tongue endurance intervention compared with the tongue strength intervention. This may have affected the reliability and consistency of the tongue endurance time scores.

Although these findings should be carefully examined because of the small sample size, the data collected in this study will add essential normative data to the database of standardized measurements for maximal strength and endurance scores in the pediatric population of the UAE. These data could be valuable for future researchers in terms of collecting additional data that can help healthcare professionals objectively evaluate children with feeding, swallowing, and speech sound production difficulties.

### Limitations and considerations for future research

Limitations should be considered when interpreting the findings of this study. (a) This is the first study to measure tongue strength and endurance in a pediatric population of Arabic language speakers, so its replication is important to confirm and validate the results. (b) This study included relatively young children who were between 3 and 8 years old. Therefore, owing to sensitive differences in ability and motivation among children in this age group, future research should consider including older children as well. (c) The global coronavirus disease 2019 pandemic caused difficulty in the recruitment of participants, particularly those with ISSDs, thereby resulting in a relatively small overall sample size ($N = 65$). This implies challenges in the generalization and clinical use of the findings of this study. Future research would thus benefit from the use of a larger sample size.

### Conclusions

This study demonstrated that tongue strength and endurance measurements were affected by age but not by sex in a sample of TD children and children with ISSDs from the UAE. It also showed that the ISSD group had lower tongue strength scores compared with the TD group. The use of the IOPI would be beneficial clinically in obtaining baseline and follow-up ongoing objective measurements of lingual strength and endurance in the pediatric population in the UAE.

### Acknowledgments

The authors would like to thank all the experts who contributed to this study.

### Author Contributions

**Conceptualization:** Mohammed Safi, Dalia Mohammad Alzyod, Maxwell Peprah Opoku, Yasser E. Agamy.

**Data curation:** Mohammed Safi, Dalia Mohammad Alzyod, Maxwell Peprah Opoku, Yasser E. Agamy.

**Formal analysis:** Maxwell Peprah Opoku.

**Investigation:** Mohammed Safi, Dalia Mohammad Alzyod, Maxwell Peprah Opoku, Yasser E. Agamy.

**Methodology:** Mohammed Safi, Dalia Mohammad Alzyod, Maxwell Peprah Opoku, Yasser E. Agamy.

**Project administration:** Mohammed Safi, Dalia Mohammad Alzyod.

**Resources:** Dalia Mohammad Alzyod.

**Writing – original draft:** Mohammed Safi, Dalia Mohammad Alzyod, Maxwell Peprah Opoku, Yasser E. Agamy.

**Writing – review & editing:** Mohammed Safi, Dalia Mohammad Alzyod, Maxwell Peprah Opoku, Yasser E. Agamy.

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
