## [Decision Letter · Decision Letter 0]

13 Jun 2023

PONE-D-23-10936Tongue Strength and Endurance among typically developing and children with idiopathic Speech Sound Disorders in the United Arab EmiratesPLOS ONE

Dear Dr. Opoku,

Thank you for submitting your manuscript to PLOS ONE. After careful consideration, we feel that it has merit but does not fully meet PLOS ONE’s publication criteria as it currently stands. Therefore, we invite you to submit a revised version of the manuscript that addresses the points raised during the review process.

ACADEMIC EDITOR:

Based on the comments from Reviewer 1 and Reviewer 2, the decision as the academic editor would be to request a major revision of the manuscript.

Reviewer 1 recommended accepting the manuscript. They found the research technically sound, supported by appropriate data and statistical analysis. They also highlighted the importance of the study in adding normative data to the database and encouraged further research in this area. No major concerns or competing interests were mentioned.

Reviewer 2, on the other hand, recommended a major revision. They acknowledged the novelty and relevance of the research but raised several issues. They pointed out that research question 2 was not fully answered in the results section, and there were inconsistencies in the writing quality, particularly in the discussion. Reviewer 2 also provided minor comments regarding clarity and phrasing in different sections of the manuscript.

Considering the major comments and the need for clarification and improvement in certain areas of the manuscript, it would be appropriate to request a major revision from the authors. They should address the concerns raised by Reviewer 2, make necessary revisions to improve the writing quality and clarity, and ensure that research question 2 is appropriately addressed in the results section. Once the revisions are made, the manuscript can be re-evaluated for potential acceptance.

. Be sure to:Indicate which changes you require for acceptance versus which changes you recommendAddress any conflicts between the reviews so that it's clear which advice the authors should followProvide specific feedback from your evaluation of the manuscriptPlease ensure that your decision is justified on PLOS ONE’s publication criteria and not, for example, on novelty or perceived impact.

We look forward to receiving your revised manuscript.

Kind regards,

Nour Shaheen

Academic Editor

PLOS ONE

Journal Requirements:

Reviewers' comments:

Reviewer's Responses to Questions

**Comments to the Author**

1. Is the manuscript technically sound, and do the data support the conclusions?

Reviewer #1: Yes

Reviewer #2: Yes

2. Has the statistical analysis been performed appropriately and rigorously? 

Reviewer #1: Yes

Reviewer #2: I Don't Know

3. Have the authors made all data underlying the findings in their manuscript fully available?

Reviewer #1: Yes

Reviewer #2: Yes

4. Is the manuscript presented in an intelligible fashion and written in standard English?

Reviewer #1: Yes

Reviewer #2: Yes

5. Review Comments to the Author

Reviewer #1: - The assessment of tongue strength and endurance and their effects on swallowing and speech forms an important component of the comprehensive evaluation of speech and language

- this study would be responsible for adding important normative data to the database of standardized measurements for maximal strength and endurance scores in the pediatric population of the United Arab Emirates. Future research is encouraged for the collection of additional data that can help healthcare professionals in objectively evaluating children with feeding, swallowing, and speech sound production difficulties.

Reviewer #2: The present manuscript describes the first normative study on tongue strength and endurance in a paediatric population in the UAE. The novelty is well described in the introduction, and the relevance of this research to clinical practise is well explained in the introduction and the conclusion. This is a small study which could have benefited from a larger sample size, as disclosed by the authors who recommended that further larger studies confirm the results. In terms of writing quality, there is some inconsistency in the style and quality throughout the manuscript; while the introduction and methods section are relatively well written, the quality declines in the discussion. this would merit reviewing for English writing to improve some parts of the manuscript. i have mentioned a couple of sentences in the minor comments.

Major comments:

- At the end of the introduction, the authors clearly stated their research questions which makes it clear for the reader what the main outcomes of the study are. However, research question number 2 is not fully answered in the results section. it appears that what was measured was the association between sex, age, and type of child, but not the association with tongue strength and endurance. If this is correct, research question 2 should not include these variables.

- In the methods section, sub-section Participants, the description of the groups is a bit confusing. it is understood that 36 TD children were included in the study and all of them were diagnosed with ISSD. See below from page 12:

"Thirty-nine TD children were recruited to participate in this study; however, three of them did not cooperate during the data collection sessions and did not therefore complete the protocol. None of them had a history....... All of them were identified as having ISSDs by a blinded SLP and were enrolled in or had undergone speech therapy. " and then the authors go on to explain in the Procedures that the participants in this group (the TD group) were recruited from rehabilitation centers. No mention here of the 29 children in the ISSD group.

- In the methods section, the fact that the instrumentation section is in between Participants description and recruitment affects the flow of the paper. It may be clearer to move the instrumentation section to after the procedure section or to move the recruitment information to the participants section.

- In the methods section, the authors should provide a rationale for their sample size calculation. Especially given the 2 groups and additional stratification of the data. Is a sample size of 65 children sufficient for normative test data? what about for group comparisons?

- In the discussion about tongue endurance, the authors argue that the lack of difference between ISSD and TD groups may be due to the difficulty explaining the procedure to the children during data collection but that their results were still consistent with other studies. However, the study they use as an example has found differences in strength with no mention here about endurance. "Nevertheless, the result is consistent with previous study which found difference between typically development and peers with speech impairment with the latter demonstrating decrease tongue strength"

Minor comments:

In the introduction, on page 10, this sentence seems to be missing an author name at the start: " found a cross-cultural difference on comparing pediatric population measures of tongue strength in Belgian children with those in American children, with the comparison revealing statistical difference on age".

In the introduction, on page 11, please correct the following sentence (repetition): "there is limited evidence on measurement of tongue strength and endurance in non– English-speaking populations is limited".

And again on page 11, repetition of the word "sex" in the following sentence:

"Is there a difference between participants in terms of child type, sex, age, sex, tongue strength, and tongue endurance?"

In the methods section, please clarify this sentence: "Because of the participants’ age, verbal motivation was given to Do you mean “encourage their participation for the continuation of the trials".

In the discussion, please rephrase: "This finding agrees with previous study conducted by which reported differences between children on age ", and correct typos "the demonstrate improve tongue"

6. PLOS authors have the option to publish the peer review history of their article (what does this mean?). If published, this will include your full peer review and any attached files.

Reviewer #1: No

Reviewer #2: **Yes: **Amira Kassis

---

## [Editor Report · Decision Letter 1]

19 Jul 2023

Tongue strength and endurance among typically developing children and children with idiopathic speech sound disorders in the United Arab Emirates

PONE-D-23-10936R1

Dear Dr. Opoku,

We’re pleased to inform you that your manuscript has been judged scientifically suitable for publication and will be formally accepted for publication once it meets all outstanding technical requirements.

Kind regards,

Nour Shaheen

Academic Editor

PLOS ONE
---

## [Editor Report · Acceptance letter]

21 Jul 2023

PONE-D-23-10936R1 

Tongue strength and endurance among typically developing children and children with idiopathic speech sound disorders in the United Arab Emirates 

Dear Dr. Opoku:

I'm pleased to inform you that your manuscript has been deemed suitable for publication in PLOS ONE. Congratulations! Your manuscript is now with our production department. 

Kind regards, 

on behalf of

Dr. Nour Shaheen 

Academic Editor

PLOS ONE